# Bone and Hemophilia: The Role of Factor VIII—Systematic Review

**DOI:** 10.3390/ijms26052172

**Published:** 2025-02-28

**Authors:** Micaela Berni, Antonella Forlino, Laura Caliogna, Liliana De Felice, Matteo Nicola Dario Di Minno, Eugenio Jannelli, Mario Mosconi, Francesca Tonelli, Camilla Torriani, Gianluigi Pasta

**Affiliations:** 1Department of Clinical, Surgical, Diagnostic and Pediatric Sciences, University of Pavia, 27100 Pavia, Italy; micaela.berni@hotmail.com (M.B.); eugenio.jannelli@unipv.it (E.J.); mario.mosconi@unipv.it (M.M.); 2Department of Molecular Medicine, Biochemistry Unit, University of Pavia, 27100 Pavia, Italy; antonella.forlino@unipv.it (A.F.); francesca.tonelli@unipv.it (F.T.); 3Orthopedics and Traumatology Clinic, IRCCS Policlinico San Matteo Foundation, 27100 Pavia, Italy; gianluigipasta@yahoo.it; 4Angelo Bianchi Bonomi Hemophilia and Thrombosis Center, IRCCS Policlinico di Milano, Ospedale Maggiore Fondation, 20122 Milano, Italy; liliana.defelice@unimi.it; 5Department of Translational Medical Sciences, Federico II University, 80131 Napoli, Italy; matteo.diminno@unina.it; 6Department of Public Health, Experimental and Forensic Medicine, University of Pavia, 27100 Pavia, Italy; camilla.torriani01@universitadipavia.it

**Keywords:** factor VIII, hemophilia, bone turnover, OPG/RANK/RANKL pathways

## Abstract

Factor VIII (FVIII) is involved in several molecular pathways and biological processes; indeed, it has a role in the coagulative cascade, cardiovascular disease, hypertension, brain and renal function, cancer incidence and spread, macrophage polarization, and angiogenesis. Hemophilic patients usually present an increase in fracture risk, bone resorption, and an excess of osteoporosis as compared to healthy individuals. Several studies have tried to clarify their etiology but unfortunately it is still unclear. This review focuses on the role of FVIII in bone biology by summarizing all the knowledge present in the literature. We carried out a systematic review of the available literature following the guidelines of the Preferred Reporting Items for Systematic Reviews and Meta-Analyses (PRISMA). Several studies demonstrated that FVIII is involved in different molecular pathways interfering with bone physiology; it exerts interesting effects on OPG/RANK/RANKL pathways and thrombin/PAR1 pathways. These data confirm a relationship between FVIII and bone metabolism; however, there are still many aspects to be clarified. This review highlights the role of the coagulation factor FVIII in bone metabolism, suggesting new hypotheses for future studies both in vitro and in vivo to better understand the important pleiotropic role of FVIII and hopefully to develop new therapeutic agents for skeletal diseases.

## 1. Introduction

Hemophilia is a rare hereditary X-linked recessive disease caused by deficient activity of coagulation factor VIII (FVIII) in hemophilia A (1:5000) and factor IX (FIX) in hemophilia B (1:30,000). Affected people show an increased bleeding risk, delayed wound healing, and frequent hemorrhage in joints that progressively leads to severe arthropathy. There are three classification degrees of hemophilia determined by the percent activity of the coagulation factor. Severe disease is defined by <1% (1 IU/dL), moderate by 1–5%, and mild by 6–40% FVIII or FIX activity.

FVIII is a plasmatic multi-domain glycoprotein (256,000 Da) that is mainly synthesized by hepatocytes but also by other tissues such as kidney, sinusoidal endothelial cell, and lymphatic tissues [1,2]. FVIII’s involvement in the coagulation cascade (cofactor facilitating the conversion of FX to its active form) is well studied, but in recent years, several scientific data have confirmed its involvement in further molecular pathways. In the bloodstream, FVIII binds in a non-covalent complex with the von Willebrand factor (VWF), a multimeric protein that is required for platelet adhesion. VWF is a chaperone and increases FVIII’s half-life and bioavailability. FVIII’s half-life is 1–2 h without VWF and 8–12 h in association with VWF. The FVIII–VWF interaction is highly dynamic, with rapid association and dissociation rates [2,3,4]. The mechanisms behind inactivated FVIII or FVIII-VWF complex clearance are not completely clarified. Recent animal studies using radiolabeled FVIII have shown that the liver, the spleen, and macrophages are the main sites were FVIII is cleared. Some in vitro and in vivo animal studies show FVIII’s involvement in several biological processes in addition to its role in the coagulative process, such as cardiovascular disease, hypertension, brain and renal function, cancer incidence and spread, macrophage polarization, angiogenesis, and bone biology [2,5,6]. FVIII’s potential roles in addition to coagulation are constantly suggested by the evidence that FVIII plasma levels are often much higher than those strictly required for the coagulation process [6].

A reduction in bone mineral density (BMD) and an increase in fracture risk have been frequently observed in hemophilic patients. Indeed, several articles show an increased rate of bone resorption and an excess of osteoporosis among these patients [1,2,7]. In recent years, in vitro and in vivo models have tried to clarify its etiology but unfortunately with very limited success [1,5,8]. Recent studies have shown a lower bone mineral density (BMD) in hemophilic patients (children and adults) compared to controls matched by age [9], and it has been shown that 70% of hemophilic patients have significantly lower BMD. Among these patients, 40% are osteopenic and 27% are osteoporotic [10,11]. The reduced BMD in people with hemophilia (PwH) has been related to multiple causes, such as reduced physical activity, recurrent hemarthroses, chronic hemophilic arthropathy, infections, and HIV or hepatitis therapy [11,12]. Only in recent studies have the molecular mechanisms through which coagulation factors may be involved in bone remodelling been evaluated [2,5]. The aim of this review is to summarize the literature evidence on the role of FVIII in bone biology.

## 2. Materials and Methods

This report provides an account of reviews in the available literature conducted in order to achieve the aim of this research. Its reporting is an adaptation of the Preferred Reporting Items for Systematic Reviews and Meta-Analyses (PRISMA) guidelines [http://www.prisma-statement.org].

### 2.1. Identifying the Research Question

The research question identified for the literature review was first of all to evaluate the molecular mechanisms through which coagulation factor VIII (FVIII) is involved in bone remodelling turnover.

### 2.2. Identifying Relevant Studies

A literature search was conducted in order to find all relevant studies on the topic. These were identified by means of diverse sources.

### 2.3. Electronic Database Search

The following electronic databases were searched, taking into consideration the chronological span between 2013 and 2023: PubMed and Scopus. The research strategy was designed to retrieve the most relevant results. Due to the specificity of the two databases employed, for each one, a different search string was built (1: PubMed search string; 2: Scopus search string). In brackets, the number of results is provided.

((“bone and bones”[MeSH Terms] OR (“bone”[All Fields] AND “bones”[All Fields]) OR “bone and bones”[All Fields] OR “bone”[All Fields]) AND (“disease”[MeSH Terms] OR “disease”[All Fields] OR “diseases”[All Fields] OR “disease s”[All Fields] OR “diseased”[All Fields]) AND (“haemophilia”[All Fields] OR “hemophilia a”[MeSH Terms] OR “hemophilia a”[All Fields] OR “hemophilia”[All Fields] OR “haemophilias”[All Fields] OR “hemophilias”[All Fields]) AND (“factor viii”[Supplementary Concept] OR “factor viii”[All Fields] OR “f8 protein human”[Supplementary Concept] OR “f8 protein human”[All Fields] OR “factor viii”[MeSH Terms] OR (“factor”[All Fields] AND “viii”[All Fields]))) AND ((english[Filter]) AND (2012:2023[pdat])) [89];AND bone AND disease AND haemophilia AND factor AND VIII AND [2012–2023]/py AND [english]/lim [92].

### 2.4. Other Sources

Three studies were also included, starting from website and citation research. These articles were regarded to be relevant, even though they were not identified through the search strings.

### 2.5. Study Inclusion Criteria

Starting from the research question, inclusion and exclusion criteria for the objective selection of the studies identified were defined. Only studies published in the English language between 2013 and 2023 were eligible for inclusion. Titles, abstracts, and full texts were screened by the research team; i.e., two authors performed the study selection and the data extraction independently, and all disagreements were discussed between the authors.

### 2.6. Data Extraction

A standardized data extraction sheet was prepared, where the main information for the studies was collected (e.g., first author’s name, study title, publication year, and DOI).

### 2.7. Study Selection

Via the literature search, seventeen studies were included in this literature review; fourteen of them were identified via database searches and three via websites or citation searching (Figure 1). Figure 1 shows the study selection process in detail, covering the number of search records retrieved from the two database searches (n = 181) and all other searches (n = 24), the number of screened titles/abstracts (n = 161), and the number of studies finally included (n = 17).

## 3. Result and Discussion

Different studies show the involvement of factor VIII in different molecular pathways related to bone physiology. The role of FVIII is evident in bone turnover, although some mechanisms are still yet to be understood.

### 3.1. OPG/RANK/RANKL Pathways in Bone Turnover

In an interesting recent study, Baud’huin et al. showed, for the first time, in in vitro models, how the FVIII-VWF complex has direct activity in osteoclast and cell apoptosis, highlighting its complex involvement in bone turnover and in bone damage associated with hemophilia and tumorigenesis. The authors demonstrate that the FVIII-VWF complex has two different ways of regulating osteoclastogenesis: first, through physical interaction with RANKL, inactivating the protein, and second, through enhancing the osteoprotegerin (OPG) effect.

RANKL (receptor activator of nuclear factor kB ligand) is a member of the tumour necrosis factor family and is mainly expressed by osteoblasts. It acts as a pro-resorption factor binding to its receptor RANK (expressed on the cell surface of osteoclast precursors), inducing osteoclastic differentiation and maturation. OPG, a soluble decoy receptor for RANK produced by osteoblasts, is an inhibitor of osteoclast differentiation, preventing the binding of RANKL to RANK and inhibiting osteoclastogenesis. The balance between these factors is tightly correlated to correct bone turnover, and any change leads to pathological conditions [2,13]. RANKL inhibition and the presence of OPG mitigate osteoclastogenesis separately, but their association increases their effect (Figure 2).

Moreover, the authors analyzed the molecular mechanisms and the interactions between RANKL, OPG, the FVIII-VWF complex, recombinant FVIII, and VWF through surface plasmon resonance (SPR), an optical technique that allows interactions between molecules to be determined. Their data show that the interaction between OPG and the FVIII-VWF complex occurred through A1 VWF’s domain and the complex FVIII-VWF, but recombinant FVIII was not able to bind to RANKL. Substantial evidence shows the anti-apoptotic properties of OPG; indeed, it is also a receptor for TRAIL (tumour necrosis factor-related apoptosis-inducing ligand), a cytokine that induces cancer cell death through apoptosis. OPG binds to TRAIL, blocking TRAIL-induced cytotoxicity, and therefore is considered a pro-tumoral agent.

Baud’huin et al. demonstrate how the FVIII-VWF complex has control over cell apoptosis; OPG does not inhibit TRAIL when the complex is present in the culture medium. The FVIII-VWF complex binds to OPG and prevents TRAIL/OPG binding by inhibiting OPG’s protective effect on apoptosis [2] (Figure 3).

Further confirmation of the involvement of FVIII in bone turnover is validated by some in vitro experiments conducted by Larson et al. They show reduced differentiation of primary mesenchymal cells (MSCs) from FVIII-deficient mice towards osteoblast lineage. Bone marrow mesenchymal stem cells from knockout (KO) mice for FVIII without replacement therapy and wild-type (WT) and KO mice for FVIII with replacement therapy were seeded and differentiated in osteoblasts. The analysis of alkaline phosphatase activity in the cultures showed that there was no significant difference between the wild-type mice and the KO mice with FVIII replacement, while the KO mice cells had reduced alkaline phosphatase activity by more than 60% [14].

An interesting recent study by Taves et al. compares bone health before and after a joint injury in KO FVIII mice (FVIII−/−) and KO VWF mice (VWF−/−). FVIII−/−, but not VWF−/−, male mice display a similarly significantly low BMD and abnormal bone structure, whereas VWF−/− mice do not show an abnormal bone phenotype. After a joint injury, FVIII−/− mice show significant trabecular bone loss, whereas VWF−/− animals do not display the same characteristics. These experimental data confirm FVIII’s involvement in the regulation of bone remodelling after injury but demonstrate how the FVIII–VWF complex is not required for bone remodelling post-injury. In addition, Taves et al. examined the serum levels of RANKL and OPG two weeks after joint injury in FVIII−/− mice. In injured FVIII−/− mice, their data show significantly lower levels of RANKL and increased expression of OPG compared to uninjured FVIII−/− controls. Injured WT mice also show a significant increase in OPG but only a mild increase in RANKL expression compared to uninjured WT. IL-6, a pro-inflammatory cytokine released locally after injury, has an effect on bone turnover independently and promotes RANKL and OPG production. It acts directly on osteoblasts to promote differentiation, but its most potent activity is on osteoclasts as a pro-resorption factor. Injury produces higher OPG/RANKL ratios in FVIII−/− mice compared to uninjured KO mice; the elevated OPG/RANKL ratio and elevated IL-6 levels are opposing forces in bone remodelling [4].

### 3.2. Role of Thrombin/PAR1 in Bone Structure

An interesting study by Aronovitch et al. evaluates the role of FVIII in hematopoietic stem cells (HSCs) in FVIII−/− KO mice. After stimulation with granulocyte colony-stimulating factor (G-CSF), they highlighted how BM shows an enhanced short-term and a reduced long-term repopulation activity and how FVIII−/− KO mice are more susceptible to G-CSF stimulation compared to WT mice. They suggested that this state could be due to the low levels of thrombin that were detected in FVIII−/− mice.

Thrombin induces the conversion of fibrinogen into fibrin, and binding to PAR-1 and PAR-4 (protease-activated receptors) expressed on the platelet membrane induces their activation and aggregation. PAR-1 and PAR-2 are expressed in osteoblast precursors, osteoblasts, and osteoclast precursors, and thrombin modulates their activities through the activation of the PAR-1 signalling pathway [3] (Figure 2). To confirm a correlation between thrombin and FVIII, the authors studied protease-activated receptor 1 knockout (PAR1 KO) and FVIII-/- KO mice, evaluating their thrombin plasma levels after stimulation with G-CSF in comparison to WT mice. Data show that thrombin levels in FVIII−/− KO and PAR-1 mice are highly reduced. These phenotypes could be mediated by the thrombin\PAR1 axis. Moreover, using micro-computed tomography (micro-CT), they evaluated bone tissue to understand the role of the FVIII/thrombin/PAR1 axis in regulating the interplay between hematopoietic stem cells (HSCs) and bone structure. Their data show similar bone abnormalities in mice of both genotypes. The serum osteocalcin (osteoblastic activity marker) quantified in FVIII KO and PAR1 KO mice compared to WT mice shows how bone loss could be associated with reduced osteoblastic activity [15] (Figure 3). However, thrombin-PAR1’s role in bone biology is still highly debated; indeed, some studies report opposite data. Tupdor et al. studied KO mice for a thrombin receptor and observed a decreasing RANKL/OPG ratio, causing increased bone mass, while Taylor et al., using small interfering siRNA to knockdown prothrombin gene expression in mice, observed that thrombin deficiency is not involved in skeletal health [7].

Goldscheitter et al. tried to identify and evaluate differences in plasma cytokine and biomarker expression related to bone metabolism between patients with hemophilia (PwH) and healthy controls. Several cytokines have direct or indirect effects on osteoblasts or osteoclasts (Figure 2). They studied the effects of lack FVIII activity on the expression of cytokines such as interferon-γ (IFN-γ), interleukin (IL-10, IL-12, IL-17a, IL1-α, IL-1ß, IL-6), M-CSF (macrophage colony-stimulating factor), TNF-α (tumour necrosis factor), RANKL, OPG, and CTX-1 (carboxy terminal telopeptide of collagen type I), a marker of bone matrix resorption. IFN-γ and TNF-α have paradoxical effects on bone, and they stimulate both osteoclast formation and bone loss. Interleukin 10 inhibits osteoclastogenesis, and IL-1α, IL-1ß, IL-6, and M-CSF regulate bone resorption by osteoclasts. The involvement of RANKL and OPG in bone turnover has been widely discussed above. All the patients had a severe FVIII deficiency (PwH); one subgroup received FVIII replacement within 24 h prior to sample collection, one subgroup received FVIII replacement on-demand, and one subgroup were on prophylactic FVIII replacement. Data were evaluated in patients both older and younger than 16 years (> or <16). No differences in either age group were found for RANKL or osteocalcin levels between the control and FVIII-deficient participants. The OPG levels in PwH ≥ 16 and those that received replacement within 24 h were high compared to their age-matched controls, but no differences were found in PwH < 16 and in patients without infusion within the prior 24 h. CTX-1 levels were elevated in PwH without infusion within the last 24 h, while CTX-1 levels in these patients were similar to the healthy controls. It is interesting to note that the CTX-1 levels in all PwH ≥ 16 were elevated compared to the controls [9].

The detectable plasma cytokines were IL-10, IL-12, and TNF-α, and these were very different between samples obtained from FVIII-deficient participants and the controls. IL-10 and IL-12 are osteoclastic inhibitory cytokines, and TNF-α is involved in bone remodelling in other disease states, although its role has not been fully clarified. The plasma levels of IL-10, IL-12, and TNF-α were significantly decreased in PwH; however, in patients who were infused within 24 h, the detection of these cytokines was similar to that in the healthy controls (Figure 3). In 2018, Haxaire et al. studied blood-induced bone loss in hemophilic mice and showed how hemophilic arthropathy (HA) is prevented by blocking inactive Rhomboid 2/a disintegrin and the metalloprotease 17/tumour necrosis factor alpha (iRhom2/ADAM17/TNF-α) pathway. Blood and its components trigger the iRhom2/ADAM17-dependent release of the pro-inflammatory cytokine TNF-α from macrophages. The Rhom2/ADAM17/TNF-α pathway is a new potential target for preventing bone resorption following joint bleeding in mice [16].

### 3.3. Scavenger Receptors in Bone Remodelling

Scavenger receptors are a family of transmembrane proteins that are used to identify several ligands and are featured in a lot of biological functions. Recently, experimental data have shown how some of these receptors are involved in the clearance of FVIII, inflammation, and bone remodelling.

Low-density lipoprotein receptor-related protein 1 (LRP-1) binds to FVIII and increases FVIII’s half-life. It is a member of the low-density lipoprotein (LDL) receptor (LDLR) family and is expressed in the liver and macrophages. LRP-1 controls osteoclast activity and consequently protects against osteoporosis. An in vitro study of macrophages with LRP-1 inactivated shows how there is a lack of positive macrophage effects in bone repair. In mice treated with recombinant LRP-1, there is an improvement in fracture healing. After evaluating the experimental data, it can be concluded that FVIII may regulate bone turnover by regulating LRP-1 expression in the cell membrane.

Scavenger receptor class A member 5 (SCARA-5) can bind to plasma VWF and/or FVIII and is expressed in a large variety of tissues, including the spleen, heart, and brain. SCARA-5 is expressed by osteoblast-like cells and inhibits their proliferation. SCARA-5 helps mesenchymal stem cells to differentiate into adipocytes; the imbalance between adipogenesis and osteoblastogenesis in favour of adipogenesis causes a BMD decrease in osteoporotic patients. These data support the involvement of FVIII and SCARA-5 in bone remodelling [3].

Recent studies have shown that factor VIII has a pleiotropic role in many physiological processes [3,6]. Bannow et al. considered several studies involving patients with hemophilia. They concluded that hemophilia may influence the risk of cardiovascular disease, with this being a risk factor for venous thromboembolism. Bannow et al. speculate that FVIII’s involvement may be linked to brain function, hypertension, renal function, and cancer incidence and spread; indeed, hemophilic patients show characteristic symptoms of these pathologies [6]. In PwH, neo-angiogenesis and an abnormal vascular architecture often occur. After joint bleeding events, markers of vascular remodelling such as CD105 and VEGF are overexpressed; this condition suggests that they may be a target of FVIII in unknown mechanisms. Moreover, VWF, which binds to FVIII, is a regulator of angiogenesis, and the enzyme that causes its degradation (ADAMTS13) has been shown to be a pro-angiogenic agent. As already mentioned above, Aronovitch et al.’s study with FVIII KO mice suggests FVIII’s potential role in hematopoiesis [15].

Recent studies have described the involvement of FVIII in macrophage function. Physiologically, monocytes differentiate into M1 (pro-inflammatory) or M2 (anti-inflammatory) macrophage subtypes depending on stimuli in the microenvironment. Monocytes differentiate into M1 macrophages in the presence of T helper cell and cytokines such as IFN-γ, TNF-α, or bacterial lipopolysaccharides (LPSs) and produce pro-inflammatory cytokines, while they differentiate into M2 macrophages in the presence of interleukins IL-10, IL-4, or IL-13 or transforming growth factor (TGF)-b. M1 has inflammatory functions, like producing pro-inflammatory cytokines, and antibacterial activity. M2 has immunoregulatory functions, increases phagocytosis, and is involved in matrix remodelling, angiogenesis, and wound healing. For the first time in the literature, Nieuwenhuizen investigated the presence of iron regulation proteins such as FPN, hepcidin, CD163, FLVCR, and HCP-1 in the human synovium, which is the main tissue involved in the most common complication of a lack of FVIII [17].

In particular, in this review, we focused on the role of factor VIII in bone physiology. Low mineral density, osteoporosis, and fractures are common comorbidities in PwH. In the last few years, the existence of a relationship between FVIII and bone metabolism has been widely confirmed; however, there are still many aspects to be clarified. In an interesting recent study using FVIII KO mice, Weitzmann et al. analyzed skeleton characteristics in both male and female animal models. Male mice have reduced trabecular bone mass in the femur and vertebrae and reduced cortical vertebral mass. Female mice only have a decrease in cortical bone in the femur and vertebrae. They observed that the mechanisms involved are different, with bone formation decreasing in males and bone resorption increasing in females. These experimental data are extremely interesting to target an appropriate therapeutic strategy to stop bone loss in patients with HA [13]. Several studies show how bone resorption in PwH is mainly due to these mechanisms of action: OPG/RANK/RANKL, thrombin-PAR1, and indirect effects due to dysregulation in cytokine profiles. Osteoporosis is a multi-factorial disease in PwH; hemarthroses and hemorrhages are usually prevented with regular factor replacement therapy, but unfortunately, prophylaxis is not always enough to avoid hemophilic comorbidities. Bone comorbidities derived from hemophilia are currently treated by recombinant coagulation factor replacement in patients as prophylaxis to prevent spontaneous bleeding. However, these therapies are not always sufficient to avoid a reduction in bone density. In a new study, Haxaire et al. observed that the steps involved in inflammation have some features in common with rheumatoid arthritis (RA). The pro-inflammatory iRhom2/ADAM17/TNF-α pathway contributes to the development of inflammatory arthritis, and the authors suggested that this pathway also contributes to the etiology of HA. They led experiments both in vitro and in vivo in a mouse model (FVIII KO) to clarify the role of this pathway in bone resorption and its correlation with HA and osteopenia. Blood and blood degradation products promote a pro-inflammatory condition by activating the iRhom2/ADAM17-dependent release of TNF-α from macrophages. The authors show how hemarthrosis promotes TNF-α production and confirm that TNF-α or iRhom2 inactivation decreases macrophage infiltration and synovitis. They show an increase in TRAP (marker for osteoclast activation) on the surface of bone next to the hemarthrosis and how a lack of TNF-α or iRhom2 mitigates bone resorption. Finally, they detect high levels of serum calprotectin in FVIII KO mice after injury, suggesting that this could be a possible hemarthrosis biomarker [16]. These data suggest a new possible therapeutic strategy for both hemophilic arthropathy and other inflammatory pathologies such as RA, aiming to inhibit TNF-α to prevent bone resorption and osteoporosis.

## 4. Conclusions

The strength of the present review rests on it being one of the few studies that highlight the relevant role of coagulation factors in bone metabolism. The aim was to shed light on actual knowledge about the molecular pathway involved and encourage new experiment studies both in vivo and in vitro to clarify the important pleiotropic role of FVIII. Also, we tried to underline new proteins and new pathways that could play a role in the development of new therapies, although there are still many aspects yet to be clarified. The main limitation of our work is that FVIII has a pleiotropic role, so it is involved in many different molecular pathways, and their dissection will require further studies to better clarify and properly understand their interactions.

## Figures and Tables

**Figure 1 ijms-26-02172-f001:**
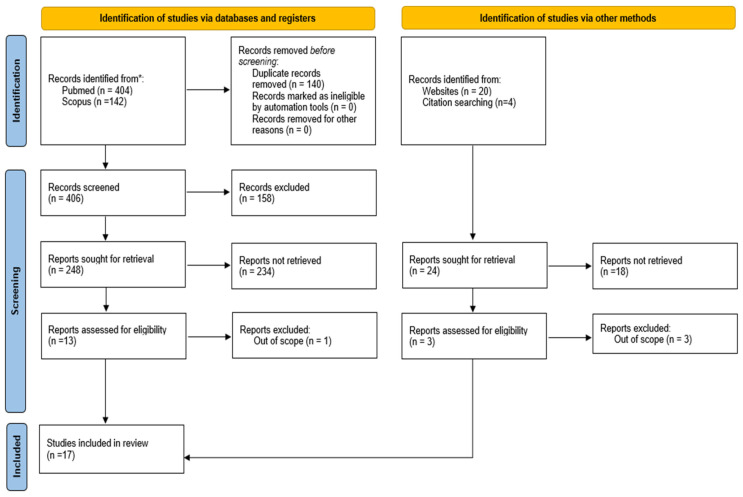
PRISMA 2020 flow diagram for new systematic reviews, which included searches of databases, registers, and other sources.

**Figure 2 ijms-26-02172-f002:**
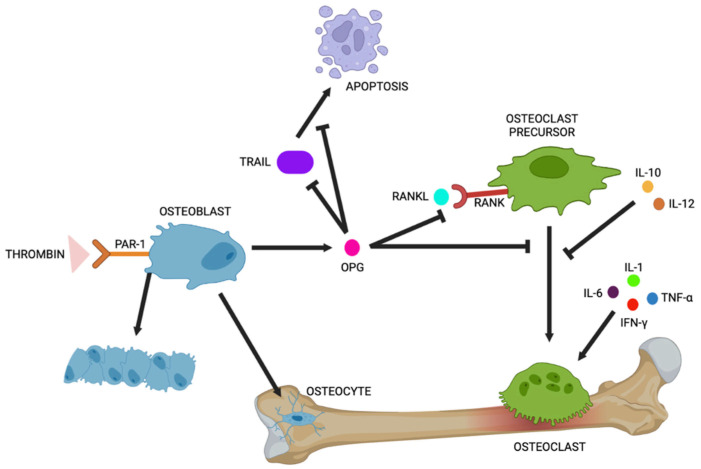
Bone turnover. The image shows the molecular pathways involved in bone turnover and the principal effects. Abbreviations: OPG (osteoprotegerin), RANK (receptor activator of nuclear factor κB), RANKL (ligand of RANK), TRAIL (TNF-related apoptosis-inducing ligand), PAR-1 (protease-activated receptors 1), IL-10 (cytokine interleukin-10), IL-12 (cytokine interleukin-12), IL-1 (cytokine interleukin-1), IL-6 (cytokine interleukin-6), IFNα (interferon gamma), TNF-α (tumour necrosis factor).

**Figure 3 ijms-26-02172-f003:**
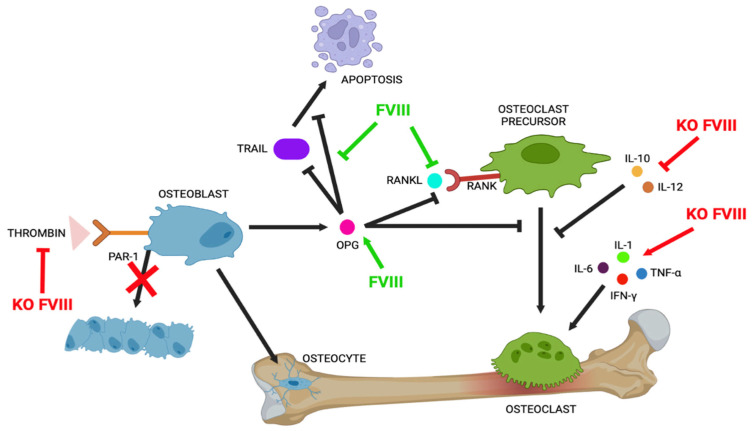
Role of FVIII in bone turnover. The image shows the role of factor VIII in bone turnover and the principal effects in the molecular pathways involve. Abbreviations: FVIII (factor VIII), OPG (osteoprotegerin), RANK (receptor activator of nuclear factor κB), RANKL (ligan of RANK), TRAIL (TNF-related apoptosis-inducing ligand), PAR-1 (protease-activated receptors 1), IL-10 (cytokine interleukin-10), IL-12 (cytokine interleukin-12), IL-1 (cytokine interleukin-1), IL-6 (cytokine interleukin-6), IFNα (interferon gamma), TNF-α (tumour necrosis factor), KO FVIII (knockout of factor VIII).

## Data Availability

The data that support the findings of this study are available within the article.

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
