# Peer review of "Bone and Hemophilia: The Role of Factor VIII—Systematic Review"

_ijms, 2025, doi:10.3390/ijms26052172_

Round 1

Reviewer 1 Report

Comments and Suggestions for Authors

It is a valuable article worth publishing, as it is focused on a very important, insufficiently clarified field with possible interferences with the therapy of haemophilia, namely the impact of different types and generations of FVIII products on the structure and physiology of bone tissue.

Proposals of small changes:

-13- introduce” main „ role….,and „ being also involved in „

-148- it would also be advisable to introduce in Figure 2 the FVIII/ FVIII-VWF to more clearly indicate  their interventions in the complex  bone turno-ver                                   

- due to the large number of new molecules, the number of abbreviations in the article  is very high ; to make the text easy understandable each abbreviation must be defined only  once, at it first usage and all abbreviations should have their  explanation: OPG (137,157,159,161 165,165,166 ,192,194...),  KO (173, 178, 178,181,185…), WT (179 ) ,iRhom 257,328,333), BMD (186, 279 );  HA is the abbreviation for haemophilia A (318) or hemophilic arthropathy (257) ?

 Corrections

18 – as compared to health patients  put  „healthy individuals”

52 - two times used the word several in a sentence.... to change with some

59 - the lack changed with the „reduction „

148 - involved

150- ligand

159- there are to joint sentences, put between them the sign ; after OPG

164- please delete ”and”

193- please add at the beginning of the sentence: in ” injured „ FVIII-/-mice

178- lineage

208- please add „in” FVIII-/-mice

284-delete it was, change them „ with being”

300- „they” before differentiate

316- delete comma before are

396- lack of data at reference13

This article is a review, which has the strength of the high number of reviewed articles and the most detailed, complex data presented. It doesn’t have weaknesses; it has no problems with patients, methods, statistical evaluations, etc. * to underline the importance of mentioning that the main role of FVIII remains its intervention in hemostasis, besides its intervention in other fields

it is essential to make more explanatory the Figure 2 * it is obligatory to explain in the text the huge number of proteins written as abbreviations involved in the complex process of FVIII action on the bone to make understandable the complex processes presented, with the numerous small (minor) changes, considering that they are important for the quality of this valuable article.

Author Response

We are grateful for your valuable comets and advices. As suggested by the reviewer we have made the following changes:

-13- introduce” main „ role….,and „ being also involved in „

Thank you very much for the advice, but the journal does not provide for the inclusion of all authors' qualifications in the affiliations

-148- it would also be advisable to introduce in Figure 2 the FVIII/ FVIII-VWF to more clearly indicate their interventions in the complex bone turnover     

Thank you very much for the advice, there are 2 images in the review, figure 2 was created to explain the mechanism of bone turnover while we created figure 3 to explain the role of fVIII in bone turnover. So you can find the request of the reviewer in figure 3. We created 2 images to better highlight the role of fVIII.

- due to the large number of new molecules, the number of abbreviations in the article  is very high ; to make the text easy understandable each abbreviation must be defined only  once, at it first usage and all abbreviations should have their  explanation: OPG (137,157,159,161 165,165,166 ,192,194...),  KO (173, 178, 178,181,185…), WT (179 ) ,iRhom 257,328,333), BMD (186, 279 );  HA is the abbreviation for haemophilia A (318) or hemophilic arthropathy (257) ?

we check all the abbreviations and correct them as suggested

18- as compared to health patients put „healthy individuals”

we correct it

52 - two times used the word several in a sentence.... to change with some

we correct it

59 - the lack changed with the „reduction „

we correct it

148 - involved

we correct it

150- ligand

we correct it

159- there are to joint sentences, put between them the sign ; after OPG

we correct it

164- please delete ” and”

we correct it

193- please add at the beginning of the sentence: in ” injured „ FVIII-/-mice

we correct it

178- lineage

we correct it

208- please add „in” FVIII-/-mice

we correct it

284-delete it was, change them „ with being”

we correct it

300- „they” before differentiate

we correct it

316- delete comma before are

we correct it

396- lack of data at reference13

we correct it

Reviewer 2 Report

Comments and Suggestions for Authors

The authors conducted a review presenting the role of FVIII in bone metabolism.

The study is well designed, but I think it should be improved.

I recommend that authors enter citations (more than one) after:  "Several in vitro and in vivo animal studies show the FVIII involvement in several biological processes in addition to its role in the coagulative process such as cardiovascular disease, hypertension, brain and renal function, cancer incidence and spread, macrophages polarization, angiogenesis, and bone biology"

Line 61 - The authors mention several articles, but only 1 is cited. 

Line 163- "Baud’huin and co-workers"- usually the citation is also included to make it easier for readers. I recommend changing with "Baud’huin et al. "The same at line 176, 184, 203. 

Please check the entire text.

Line 282- What are these recent studies?

Author Response

We are grateful for your valuable comets and advices. As suggested by the reviewers we have made the following changes:

- I recommend that authors enter citations (more than one) after:  "Several in vitro and in vivo animal studies show the FVIII involvement in several biological processes in addition to its role in the coagulative process such as cardiovascular disease, hypertension, brain and renal function, cancer incidence and spread, macrophages polarization, angiogenesis, and bone biology"

Thank you very much for the advice, we do it.

- Line 61 - The authors mention several articles, but only 1 is cited.

We add the citations of other articles.

Line 163- "Baud’huin and co-workers"- usually the citation is also included to make it easier for readers. I recommend changing with "Baud’huin et al. "The same at line 176, 184, 203. Please check the entire text.

Thank you very much for the advice, we do it.

Line 282- What are these recent studies?

Thank you very much for the advice, we add the citations of these studies.

Round 2

Reviewer 2 Report

Comments and Suggestions for Authors

Line 59- bone density (BMD), I recommend writing bone mineral density

Author Response

We are grateful for your valuable comets and advices.

As suggested by the reviewer we have made the following changes:

Line 59- bone density (BMD), I recommend writing bone mineral density

we do it.